



# Revisiting the reaction of dicarbonyls in aerosol proxy solutions containing ammonia: the case of butenedial

Jack C. Hensley[1], Adam W. Birdsall[2,a], Gregory Valtierra[3], Joshua L. Cox[2], Frank N. Keutsch[1,2,4]

[1]School of Engineering and Applied Sciences, Harvard University, Cambridge, MA, USA
[2]Department of Chemistry and Chemical Biology, Harvard University, Cambridge, MA, USA
[3]Harvard College, Cambridge, MA, USA
[4]Department of Earth and Planetary Sciences, Harvard University, Cambridge, MA, USA
[a]now at: Goodyear, Akron, OH, USA

*Correspondence to*: Jack C. Hensley (jhensley@g.harvard.edu) and Frank N. Keutsch (keutsch@seas.harvard.edu)

**Abstract.** Reactions in aqueous solutions containing dicarbonyls (especially the α-dicarbonyls methylglyoxal, glyoxal, and biacetyl) and reduced nitrogen ($NH_X$) have been studied extensively. It has been proposed that accretion reactions from dicarbonyls and $NH_X$ could be a source of particulate matter and brown carbon in the atmosphere and therefore have direct implications for human health and climate. Other dicarbonyls, such as the 1,4-unsaturated dialdehyde butenedial, are also produced from the atmospheric oxidation of volatile organic compounds, especially aromatics and furans, but their aqueous

phase reactions with $NH_X$ have not been characterized. In this work, we determine a pH-dependent mechanism of butenedial reactions in aqueous solutions with $NH_X$ that is compared to α-dicarbonyls, in particular the dialdehyde glyoxal. Similar to glyoxal, butenedial is strongly hydrated in aqueous solutions. Butenedial reaction with $NH_X$ also produces nitrogen-containing rings and leads to accretion reactions that form brown carbon. Despite glyoxal and butenedial both being dialdehydes, butenedial is observed to have three significant differences in its chemical behavior: (1) as previously shown, butenedial does

not substantially form acetal oligomers, (2) the butenedial/$OH^-$ reaction leads to light-absorbing compounds, and (3) the butenedial/$NH_X$ reaction is fast and first order in the dialdehyde. Building off of a complementary study on butenedial gas-particle partitioning, we suggest that the behavior of other reactive dialdehydes and dicarbonyls may not always be adequately predicted by α-dicarbonyls, even though their dominant functionalities are closely related. The carbon skeleton (e.g., its hydrophobicity, length, and bond structure) also governs the fate and climate-relevant properties of dicarbonyls in the

atmosphere. If other dicarbonyls behave like butenedial, their reaction with $NH_X$ could constitute a regional source of brown carbon to the atmosphere.

## 1 Introduction

Atmospheric particles are known to contain organic compounds that absorb light (Andreae and Gelencser, 2006; Bond et al., 2007; Laskin et al., 2015). One source of this so-called "brown carbon" is the irreversible reaction between dicarbonyls and

reduced nitrogen ($NH_X$, here $NH_X = NH_3 + NH_4^+$) that takes place in the aqueous phase of atmospheric particles and in cloud





droplets (Debus, 1858; Loeffler et al., 2006; McNeill, 2015; Nozière et al., 2007; Volkamer et al., 2007). Because the ensuing accretion reactions form low-volatility products that tend to remain in the particle phase, brown carbon chemistry increases both the loading of atmospheric particles and their capacity to absorb light. Thus, aqueous-phase dicarbonyl/NH$_X$ reactions in the atmosphere have direct implications for human health and climate (Kanakidou et al., 2005; Pöschl, 2005).

35          Members of the dicarbonyl family are characterized by their dominant functionality, which is the identity of their two carbonyls. Dicarbonyls are either diketones, e.g., biacetyl, dialdehydes, e.g., glyoxal and butenedial, or ketoaldehydes, e.g., methylglyoxal. The number of aldehydes versus ketones has a strong influence on chemical reactivity. Atmospheric dicarbonyls can have long or short carbon backbones that are saturated or unsaturated (Bierbach et al., 1994; Obermeyer et al., 2009). The most measured and studied dicarbonyls in the atmosphere are the α-dicarbonyls, in particular glyoxal ($C_2H_2O_2$)

and methylglyoxal ($C_3H_4O_2$), which are the smallest dialdehyde and ketoaldehyde, respectively. Recently, biacetyl ($C_4H_6O_2$), the smallest diketone, has also received scientific attention (Grace et al., 2020; Kampf et al., 2016). Larger, complex dicarbonyls are also thought to be important products of biomass burning and fossil fuel combustion (Arey et al., 2009; Aschmann et al., 2011, 2014; Gómez Alvarez et al., 2007, 2009; Volkamer et al., 2001; Yuan et al., 2017), but they have rarely been studied or quantified in the atmosphere. Challenges include their high reactivity, no commercial availability, and

difficulties in analysis via chromatography (Hamilton et al., 2003; Smith et al., 1999). However, if the dominant functionality governs chemical behavior, then the well-understood α-dicarbonyls can "stand in" for others from the larger compound class, but it is important to evaluate the limitations of this approach.

          Quantitative understanding of dicarbonyl/NH$_X$ chemistry has come from laboratory studies of glyoxal, methylglyoxal, and biacetyl in bulk aqueous solutions that contain ammonium salts like ammonium sulfate (AS) (Grace et al., 2020; Kampf

et al., 2016; Nozière et al., 2009; Sareen et al., 2010; Yu et al., 2011), and to an extent from aerosol chamber studies (De Haan et al., 2017, 2018; Galloway et al., 2009). Bulk solutions can mimic the aqueous phase of actual atmospheric particles, albeit at lower ionic strength, and have the advantage that established organic analysis methods like nuclear magnetic resonance (NMR), mass spectrometry (MS), and UV/Vis spectroscopy can be used to identify reaction products and quantify reaction rates. The reaction mechanisms can inform chemical models that estimate for example the warming effect of brown carbon on

climate (Ervens, 2015; Saleh, 2020).

          When glyoxal, methylglyoxal, or biacetyl and ammonium salts are introduced to an aqueous mixture, the medium darkens. The color change has been attributed to accretion products that in spite of their low abundance can absorb light effectively (Grace et al., 2020; Kampf et al., 2012; Nozière et al., 2007; Shapiro et al., 2009; Yu et al., 2011). Irreversible reaction with NH$_X$ produces nitrogen-containing unsaturated rings, e.g., imidazoles in the glyoxal/NH$_X$ reaction, and is the

starting point for further accretion reactions. Yu et al. (2011) and Kampf et al. (2012) demonstrated that imidazole formation from glyoxal/NH$_X$ follows a rate law of the form k[GL]$^2$[NH$_4^+$][NH$_3$], and is as such fastest at neutral to basic pH (Maxut, 2015). Although these products have also been detected in chamber experiments of deliquesced AS aerosol, the reaction is observed to be too slow at typical dicarbonyl concentrations and aerosol pH and NH$_X$ to affect chemical composition, in part due to the quadratic dependence on glyoxal (Galloway et al., 2009; Yu et al., 2011). However, glyoxal/NH$_X$ reactions have



been shown to be greatly accelerated in evaporating cloud droplets (Lee et al., 2013) and produce strongly absorbing compounds even at low concentrations (Shapiro et al., 2009). Methylglyoxal/$NH_X$ reactions in bulk solutions are faster and are shown to be linearly dependent on methylglyoxal and ammonium, but they too may be limited in the atmosphere (Powelson et al., 2014; Sareen et al., 2010; Sedehi et al., 2013). Biacetyl/$NH_X$ reactions are also shown to be linearly dependent on the dicarbonyl and ammonium and can form light-absorbing species, although biacetyl is much less hydratable and is not as

relevant to aqueous aerosol (Grace et al., 2020). Further supported by field observations, modeling, and other laboratory work, the consensus is that dicarbonyl/$NH_X$ reactions probably are too slow to contribute substantial particle mass in the atmosphere (Ervens and Volkamer, 2010; Laskin et al., 2015).

Beyond reaction with $NH_X$, past work with α-dicarbonyls suggests that the dicarbonyl family tends to behave similarly in aqueous solutions. At basic pH, methylglyoxal and glyoxal both react with $OH^-$ to form colorless oligomeric

products. Both are hydratable when dissolved in an aqueous solution, and therefore have suppressed vapor pressures over aqueous media. However, there are key differences in the behavior of these compounds, which have been attributed to the dicarbonyl moiety. Glyoxal favors a dihydrate state in dilute solutions that can oligomerize with mono- or unhydrated glyoxal to form cyclic hemiacetals. Methylglyoxal hydration occurs predominantly at the aldehyde, resulting in a ketodiol that is much less reactive than glyoxal's monohydrate, which possesses an aldehydic group. Oligomerization of methylglyoxal and biacetyl

proceeds predominantly through aldol condensation rather than through the formation of acetals. In the presence of anions like sulfate, methylglyoxal vapor pressure is increased (salting out) whereas glyoxal's decreases (salting in) (Kampf et al., 2013; Wang et al., 2014; Waxman et al., 2015). The accumulation of such dissimilarities can lead to different overall reactivities and solubilities between glyoxal and methylglyoxal. For example, glyoxal has an effective Henry's Law constant that can be ~4 orders of magnitude larger than methylglyoxal when partitioning over salt-containing solutions (McNeill, 2015; Waxman et

al., 2015), which can impact the loading and composition of atmospheric aerosol. The Henry's law constant of biacetyl is even lower than that of methylglyoxal. In general, it is thought that behavioral differences can be explained by the ketone versus aldehyde content of dicarbonyls and, thus, butenedial would be expected to behave similarly to glyoxal.

The influence of carbon backbone structure on dicarbonyl reactivity has received far less attention, including in reactions with $NH_X$. Kampf et al. (2016) observed that the reactions of two larger dicarbonyls, 2,5-hexadione (an unsaturated

1,4-diketone) and glutaraldehyde (an unsaturated 1,5-dialdehyde), with $NH_X$ are fast, and their products may be more light-absorbing than those of α-dicarbonyls/$NH_X$ reactions. Building off of this work, we investigate butenedial reactions in aqueous solutions containing $NH_X$ and compare them to those of α-dicarbonyls. Butenedial is an unsaturated 1,4-dialdehyde observed as a major product of aromatic and furan degradation in laboratory studies (Arey et al., 2009; Aschmann et al., 2014; Bierbach et al., 1994; Coggon et al., 2019; Stockwell et al., 2015; Strollo and Ziemann, 2013; Volkamer et al., 2001) and in ambient air

(Obermeyer et al., 2009). As butenedial is an electron-poor dialdehyde, its chemical reactivity is expected to be more similar to that of glyoxal than methylglyoxal or biacetyl. However, we have shown in a past study that butenedial acetal oligomers are insignificant in water and that it salts out in the presence of anions, behavior that is unlike that of glyoxal and more similar to methylglyoxal (Birdsall et al., 2019).





Butenedial is studied first in aqueous solutions without $NH_X$, in which it reacts reversibly with $H_2O$ and irreversibly
with $OH^-$. Then it is studied in AS aqueous solutions, in which it can additionally react irreversibly with $NH_X$. Reaction
products and reaction rates are observed with NMR to characterize the chemical mechanism. The reactivity of butenedial is
compared to glyoxal as well as methylglyoxal and biacetyl. Implications for unsaturated 1,4- and other dicarbonyls are
considered. The atmospheric relevance of dicarbonyl/$NH_X$ reactions is re-examined, including as a source of brown carbon.

## 2 Methods

A chemical mechanism was developed for butenedial in an aqueous $NH_X$ solution. Butenedial reaction with $OH^-$ or $NH_X$ was
studied in bulk solutions with conditions given in Table 1. As is justified later, reaction with $OH^-$ was found to be negligible
at the pH and $NH_X$ conditions of the butenedial/$NH_X$ reaction studies, and vice versa, as no overlapping products were observed
in solutions. No other reactions were observed. Measurements of the composition of the bulk solutions were taken with NMR
or MS and identified reaction products were used to formulate a chemical scheme. A custom kinetic mechanism was fit to the
temporal evolutions of reactants and reaction products (via NMR only), from which rate constants were empirically
determined. Predictions with the kinetic mechanism were compared against additional experimental measurements at different
pH and $NH_X$.

### 2.1 Materials and instruments

All chemicals were obtained from Sigma Aldrich unless otherwise specified. The synthesis of butenedial is described
elsewhere in greater detail (Avenati and Vogel, 1982; Birdsall et al., 2019). Mixtures containing 2.4 M 2,5-dihydro 2,5-
dimethoxyfuran (TCI America, 98%) and 3.4 M glacial acetic acid (HAc, VWR, 99.7%) were prepared. After about 10 days
of reaction at room temperature, we purified mixtures with rotary evaporation to 75% butenedial by weight (w/w). The
remaining 25% was predominantly residual water and HAc. All mixtures were well mixed at the start of reactions. Reacting
mixtures were kept in capped glass vials or capped NMR tubes without further stirring.

All 1D 1H-NMR spectra were collected with Varian 400 MHz spectrometers, 64 scans per recorded spectrum.
Deuterium oxide ($D_2O$, 99.9 atom % D) was the solvent for NMR experiments. The methods described by Yu et al. (2011)
were followed to estimate molarities quantitatively in 1H-NMR spectra with 1% w/w dimethyl sulfone (DMS, 99%). The pH
was estimated from the spectra by tracking the proton shift of an acid near 50% dissociation, according to the methods
described by Yu et al. (2011). The acid that was tracked depended on the solution pH: if solution pH 3-6, residual acetic acid
was tracked; if solution pH 6-8.5, 1% w/w methylphosphonic acid (MPA, 98%) was added and tracked; and if solution pH 9-
11, 1% w/w 2,4,6-trimethylphenol (TMP, 99%) was added and tracked.

MS spectra were collected of butenedial/$OH^-$ mixtures with a liquid chromatography – mass spectrometry (LC-MS)
and of butenedial/$NH_X$ mixtures with a commercial time-of-flight mass spectrometer (TOF-MS, JEOL AccuTOF). Birdsall et
al. (2018) describes the operation of the TOF-MS described in detail. ~140 pL aqueous particles were produced from the



analyte solution with a droplet-on-demand injector. The particles were briefly (< 1 s) in contact with dry nitrogen gas. Reaction and evaporation of solutes were negligible while the particles traveled from injector to a glass slide heated at 220 °C. The glass slide heated the particle, and a corona discharge ionized the resulting vapors to be drawn into the inlet of the TOF-MS. Mass spectral signals were recorded as counts per integer mass-to-charge (m/z) channel. 10-20% w/w hexaethylene glycol (PEG-6, 99%) was used as an internal standard for MS measurements, as done previously (Birdsall et al., 2018). Previous work

demonstrated there is no reaction between butenedial and PEG-6 (Birdsall et al., 2019). Distilled $H_2O$ was the solvent for all MS experiments. The TOF-MS was operated in positive mode while the LC-MS was operated in negative mode.

## 2.2 Chemical analysis

### 2.2.1 Butenedial in aqueous solutions without $NH_X$

0.55 M butenedial in $D_2O$ solution and a six-fold dilution of this solution, resulting in 0.09 M butenedial in $D_2O$, were studied

with NMR under acidic conditions. In both solutions, butenedial strongly favored the dihydrate form with only minimal formation of acetal oligomers (see Supplement Section 2, Figure S2 and Table S2), as shown previously (Birdsall et al., 2019). Additionally, four solutions containing 0.2-0.3 M butenedial were prepared in $D_2O$ buffered with 1 M $Na_2CO_3$-$NaHCO_3$ to pH 8.8-10.4. All solutions immediately turned dark brown. Butenedial was measured quantitatively with NMR throughout its reaction with $OH^-$. Disproportionation products were observed (Figure S4), including the growth of broad peaks embedded in

the baseline that were indicative of accretion reactions.

### 2.2.2 Butenedial in aqueous solutions with $NH_X$

0.9 M butenedial/0.45 M AS (VWR, > 99%) mixtures were prepared in water and $D_2O$ with the internal standards PEG-6 or DMS and 0.5 M sodium carbonate ($Na_2CO_3$) – sodium bicarbonate ($NaHCO_3$) buffer. The solution immediately turned orange brown. After 20 min of reaction, mass spectra of the mixtures indicated nitrogen-containing products with signals at m/z 84,

149, 150, and 168, assumed to be adducts with $H^+$ (Figure S7). The most reasonable chemical formulas of these products were $C_4H_5NO$ (83 Da), $C_8H_8N_2O$ (148 Da), $C_8H_7NO_2$ (149 Da), and $C_8H_9NO_3$ (167 Da). $C_8H_9NO_3$ was the parent molecule for the $C_8H_7NO_2$ fragment.

The 1H-NMR spectra (Figure S8) showed distinct groups of quantitative related signals that had similar temporal behavior. Each group of peaks whose quantitative signal strength behaved as integers and had the same temporal behavior was

presumed to arise from a single compound. A molecular structure was proposed for each cluster of peaks and the molecular formulas mentioned above. The inferred products were as follows: 2-pyrrolinone (pyrrolinone, PR, $C_4H_5NO$), and a butenedial-pyrrolinone "dimer" (BD-PR, $C_8H_9NO_3$). We propose that 2-butenal-1,3-diazepine (diazepine, DZ, $C_8H_8N_2O$) is a minor product that is observable with MS but was not detected with the less sensitive NMR. The growth of broad peaks embedded in the baseline suggested substantial formation of N-containing accretion products that were likely strongly π-conjugated and

should therefore absorb light and could explain the dark color of the solution.





### 2.3 Kinetic mechanism

The kinetic mechanism was formulated as a system of ordinary differential equations, one ordinary differential equation per identified chemical compound. The butenedial reaction with $OH^-$ was quantified and used as foundation for the mechanism with $NH_X$. Rate laws were formulated based on known reactions of related species and then adjusted to optimize agreement

between the mechanism and observations (see Section 3). Python's scipy package was used to parameterize each fit rate constant's mean value and standard error. The lmfit library with the Levenberg-Marquardt algorithm was used to perform the least squares minimization. A Monte Carlo simulation was performed to derive the reported 95% confidence intervals on model runs.

### 2.3.1 Butenedial/$OH^-$ reaction

Dicarbonyl/$OH^-$ reactions are known to be effectively irreversible and are characterized by a rate law that is first order in the dicarbonyl (R1). According to Fratzke & Reilly (1986), the dicarbonyl/$OH^-$ reaction rate constant ($k_1$) is a function of $OH^-$ as defined by the following relationship:

$$k_1 = \frac{a_I[OH^-] + a_{II}[OH^-]^2}{1 + a_{III}[OH^-]},$$ (1)

where $a_I$ and $a_{II}$ are related to the role of the hydrated anion or dianion, respectively (Fratzke and Reilly, 1986). The coefficients

$a_I$, $a_{II}$, and $a_{III}$ were fit with four unique $[OH^-]/k_1$ pairings, each corresponding to a different experimental run. $k_1$ was derived from the first order loss of butenedial in each experiment (Figure S12). Subsequently, $a_I$-$a_{III}$ were determined using the scipy optimize.curve_fit library implemented with the Trust Region Reflective minimization method.

### 2.3.1 Butenedial/$NH_X$ reaction

A system of three ordinary differential equations (DE1-DE3) was used to model butenedial, pyrrolinone, and butenedial-

pyrrolinone dimer concentrations, with pH and initial concentrations of reactants and products as inputs. Rate constants for five rate laws (R2-R6) were fit to experimental data, with starting conditions of 0.9 M butenedial and 0.9 M $NH_X$, and pH ranging 4.2-5.7. $k_1$ was implemented according to the fitting described in Section 2.3.1. pH was estimated with an empirical formulation that agreed closely with measurements (Figure S13). Model performance was assessed against measurements taken in bulk liquid experiments with different initial conditions and pH (Figures S15-S16). One limitation was that reaction

rates of unmeasured species had to be approximated with a proxy, i.e., in the cases of $NH_X$ and accretion product concentrations. As is discussed in Section 3, the approximations have minimal impact on the prediction of butenedial loss and on the estimation of most parameters.



## 3 Results

A chemical scheme for butenedial in an aqueous solution with $NH_X$ is proposed in Section 3.1. Butenedial reaction with $OH^-$
and reaction with $NH_X$ are described in Section 3.2 and Section 3.3, respectively. The fate of butenedial in aqueous solutions
with and without $NH_X$ is summarized in Section 3.4.

### 3.1 Chemical scheme

Butenedial in aqueous $NH_X$ solutions of pH 3.6-10.4 is proposed to obey the chemical scheme shown in Figure 1, which
demonstrates that it can undergo three reactions. First, butenedial can be reversibly hydrated and is observed to prefer the
dihydrate form in aqueous solutions without evidence for significant acetal oligomer formation. Second, butenedial reacts with
$OH^-$ to form irreversible reaction products such as a hydroxy acid, which ultimately lead to oligomeric, light-absorbing
compounds. Third, butenedial reacts with $NH_3$ to produce an imine intermediate which forms irreversible reaction products
(pyrrolinone, a diazepine, and a butenedial-pyrrolinone "dimer") and also ultimately lead to oligomeric, light-absorbing
compounds. These accretion products are observed to be reactive with butenedial, pyrrolinone, and the butenedial-pyrrolinone
dimer.

As discussed in a previous study, like glyoxal, the hydration equilibrium of butenedial is strongly shifted toward the
dihydrate (>95% of the total butenedial on a molar basis). Birdsall et al. (2019) also showed that the ratio of unhydrated or
monohydrated to dihydrated butenedial appeared to be unaffected by the availability of water. It is expected that the hydration
behavior of butenedial will affect its reactivity, and possibly differentiate it from glyoxal, which has been observed to form a
highly reactive yet soluble monohydrate form. Additionally, in contrast to glyoxal, which readily forms glyoxal acetal
oligomers, no butenedial acetal oligomers were observed by Birdsall et al. (2019). Our experiments support that only minimal
acetal oligomer formation is possible and is much less pronounced than for glyoxal. This behavior is not typical for dialdehydes
but has been observed for adipaldehyde (Hardy et al., 1972). One explanation is that butenedial, like adipaldehyde, has a
hydrophobic center that influences the ability of its hydrates to oligomerize like glyoxal.
The 1H-NMR spectra of butenedial in basic aqueous solutions and solutions with $NH_X$ both exhibit the buildup of
signal in the baseline in the vicinity of possible monomer peaks (shifted to lower ppm, Figure S2, S5). This buildup increases
in intensity and spreads out with respect to chemical shift over time. Thus, accretion reactions take place that are the ultimate
sink for butenedial and its reaction products, producing low-volatility compounds that can explain the brown color. Reaction
products of dicarbonyl/$OH^-$ reactions are not thought to be reactive to the dicarbonyl or to products of dicarbonyl/$NH_X$
reactions, such as in the case of glyoxal (Yu et al., 2011). Butenedial loss is observed to be first order in butenedial at all time
scales and [$OH^-$], indicating that butenedial/$OH^-$ reaction does not result in additional butenedial removal from solution, e.g.,
via products reacting with butenedial. On the other hand, butenedial is reactive with butenedial/$NH_X$ reaction products, as has
been observed in analogous reactions of glyoxal, methylglyoxal, and biacetyl, and they and higher accretion products increase
butenedial loss.



## 3.2 Butenedial/OH⁻ reaction

Estimates of the a-coefficients of $k_1$ are shown in Table 2. The dependence of the pseudo first-order rate constant $k_1$ on OH⁻ is shown in Figure 2. The rate constant is $< 1\times10^{-4}$ s⁻¹ at pH < 9, indicating that butenedial/OH⁻ reaction is insignificant except at basic pH. No evidence of reaction has been observed in standard butenedial solutions (pH ~ 4) that we have kept on the shelf for months. At solution pH 8.5-8.8, butenedial loss from butenedial/OH⁻ reaction is negligible compared to butenedial/NH$_X$ reaction (Figure S14). We therefore conclude that butenedial/OH⁻ reaction is insignificant at neutral and acidic conditions relevant to the atmosphere, especially when NH$_X$ is present, although the parametrization is included in the butenedial/NH$_X$ kinetic mechanism.

## 3.3 Butenedial/NH$_X$ reaction

Five chemical reactions (R2-R6) explain the evolution of butenedial (BD) and its major reaction products with NH$_X$, pyrrolinone (PR) and butenedial-pyrrolinone "dimer" (BD-PR). Table 3 shows the proposed reactions, their rate laws and fitted rate constants. The reactions and their rate laws are discussed here and the fitted rate constants in Section 3.3.2.

The initial, irreversible reaction between butenedial and NH$_3$ (R2) is linearly dependent on both species and produces pyrrolinone. While the reaction is acid catalyzed, the rate constant of dialdehyde/NH$_X$ reactions is pH dependent (Yu et al., 2011), resulting in a pH-independent rate law. At constant NH$_3$, the reaction is pseudo first order in butenedial. In analogy to the related glyoxal and methylglyoxal reactions, we propose an imine intermediate for this reaction. Reaction of pyrrolinone with butenedial is pH-dependent and produces a butenedial-pyrrolinone dimer (R3). We anticipate that the proposed imine can also undergo an acid-catalyzed reaction to produce a diazepine (DZ).

Reactions between each of butenedial, pyrrolinone, butenedial-pyrrolinone dimer and an accretion product term (R4-R6) are included to represent the removal of these species through accretion reactions. Accretion reactions have been observed in studies with glyoxal (Kampf et al., 2012; Yu et al., 2011). These reactions typically involve oligomer-like molecules made up of precursor compounds (in this case, butenedial), its reaction products (pyrrolinone and butenedial-pyrrolinone dimer), and products of subsequent reactions. The resulting accretion products are diverse, as is known for similar chemical systems, and were not quantified directly with 1H-NMR. Therefore, to include these reactions in the kinetic mechanism, the accretion product (AP) concentration is approximated with butenedial-pyrrolinone dimer as a proxy: [AP] = [BD-PR]. Setting the AP concentration equal to BD-PR concentration involves several assumptions, namely that the number of AP reactive sites scales with BD-PR concentration, the molecular weight distribution of AP members is independent of pH, and any reversibility in accretion reactions can be accounted for with this approximation. However, strong agreement is still observed between butenedial observations and model predictions under different pH and initial reactant conditions, which suggests that this approximation does not significantly affect mechanistic portrayal of butenedial reactivity.

$$\frac{d[BD]}{dt} = -k_1[BD] - k_2[BD][NH_3] - k_3[BD][PR][OH^-] - 2\,k_4[BD][AP] \tag{DE1}$$




$$\frac{d[PR]}{dt} = k_2[BD][NH_3] - k_3[BD][PR][OH^-] - k_5[PR][AP] \tag{DE2}$$

$$\frac{d[BD\text{-}PR]}{dt} = k_3[BD][PR][OH^-] - k_6[BD\text{-}PR][AP] \tag{DE3}$$

255    The butenedial/$NH_X$ kinetic mechanism contains three differential equations (DE1-DE3), one per explicitly measured species: butenedial (BD), pyrrolinone (PR), and butenedial-pyrrolinone dimer (BD-PR). The finalized mechanism contains the best fit reaction rate constants to experimental data; the resulting model output and experimental data are shown in Figure 3. The mechanism correctly captures the evolution of each species across the wide range of pH and reactant concentrations in the experiment, which are especially relevant to those typical of atmospheric particles.

260    pH is modeled independently with an empirically derived fit. The fitting of pH to the empirical law causes a maximum deviation of 0.1 pH units from measured pH (Figure S13). For the fitting, $NH_X$ is not measured and instead is estimated with the following relationship: $[NH_X] = [NH_{X,0}] - ([BD_0] - [BD])$, i.e., one $NH_X$ is lost per butenedial. This simplification artificially consumes $NH_X$ when butenedial dimerizes with pyrrolinone or accretes, and $NH_X$ loss could be overestimated. To assess the sensitivity of the parametrization to $NH_X$, the model fitting was also performed assuming $NH_X$ is not consumed during reaction (i.e., zero $NH_X$ is lost per burtenedial). The parameter fitting with this scenario provides a maximum deviation as it is effectively the opposite extreme. The differences between these produced parameters and those of the original fitting were small, typically falling <5% of the parameter estimates with one exception (Figure S14). The employed simplification of $NH_X$ is therefore assumed to have minor effects on the kinetic mechanism.

    Model predictions using this mechanism compare well with measurements from two additional experiments with different pH and initial conditions (Figure S15-S16). One was performed with 0.4 M $BD_0$, 0.4 M $NH_{X,0}$, and pH 3.6 and the other with 0.9 M $BD_0$, 0.2 M $NH_{X,0}$, and pH 8.5-8.8. This indicates that the kinetic mechanism is robust across a relevant range of pH and initial conditions.

**3.4 Comparison of butenedial loss processes in aqueous aerosols**

The lifetime of condensed-phase butenedial from aqueous reaction with $OH^-$ or $NH_X$, with AS as $NH_X$ source, is compared to that of wet deposition, which for tropospheric aqueous particles is about one week (Seinfeld and Pandis, 2016). The dominant first order loss process of butenedial is shown as a function of $NH_X$ and pH in Figure 4. Butenedial/$OH^-$ is the main sink if < 1 mM $NH_X$ and above ~pH 7, although this pH is not particularly atmospherically relevant. Reaction with $NH_X$ can be fast at typical $NH_X$, even under somewhat acidic conditions, and is therefore competitive with wet deposition. Butenedial loss is increased through accretion reactions in the $NH_X$ pathway but this effect is not included in this analysis. Thus, the figure represents a lower limit for butenedial loss via the $NH_X$ pathway.





## 4 Discussion

### 4.1 Comparison of dicarbonyl reactivity in aqueous $NH_X$ solutions

The reactivity of dicarbonyls in aqueous $NH_X$ solutions was previously understood primarily on the basis of the dominant functionality through studies of α-dicarbonyls. Following the work of Kampf et al. (2016), this study provides an additional perspective that considers the role of the carbon skeleton on dicarbonyl behavior. Biacetyl is the least hydratable of the α-dicarbonyls, and because it is a diketone, is the least similar to butenedial. Methylglyoxal is less hydrated than glyoxal and butenedial because it is a ketoaldehyde that has a hydrophobic methyl moiety. Butenedial is an electron-poor dialdehyde like glyoxal and is therefore strongly hydrated, but it also has a hydrophobic alkene group. In addition, it has a four-carbon chain, making the formation of stable five-membered organic molecules thermodynamically and especially kinetically favorable in comparison to glyoxal, methylglyoxal, and biacetyl, which provides evidence for the importance of the carbon backbone to dicarbonyl chemistry.

The dicarbonyl moiety leads to several similarities in the behavior of butenedial, glyoxal, methylglyoxal, and, to a lesser degree, biacetyl in aqueous $NH_X$ solutions: all hydrate reversibly, react with $OH^-$ under basic conditions, and react with $NH_X$ to produce heterocycles and subsequently undergo accretion reactions. Brown carbon is produced in solution even if minimal reaction has occurred and accelerates with increasing pH. However, we demonstrate three important differences between butenedial and glyoxal in particular that showcase the variability possible between dicarbonyls and even electron-poor dialdehydes.

First, unlike butenedial, glyoxal shows a strong tendency to form acetal oligomers in pure aqueous solutions. Biacetyl and methylglyoxal can also form acetal oligomers in aqueous solutions, although aldol condensation products are more common (Grace et al., 2020; Sareen et al., 2010). Somewhat surprisingly, butenedial acetal oligomers are much less pronounced, despite butenedial having two reactive aldehyde groups and being predominantly dihydrated in aqueous solution. This behavior was reported by Birdsall et al. (2019) who did not find any acetal products from butenedial itself or, perhaps even more surprisingly, in the presence of high concentrations of polyethylene glycol. One explanation could be that butenedial, like methylglyoxal and biacetyl, has a substantial hydrophobic component that influences the ability of its hydrates to oligomerize like glyoxal, which was not expected based on the similarity with glyoxal in hydration behavior.

It is also observed that the double bond within the carbon backbone affects the properties of the products. Polymers of lactic acid and glycolic acid (the products of methylglyoxal and glyoxal reaction with $OH^-$) are colorless, presumably because they lack π-conjugated double bonds. Accretion products from butenedial/$OH^-$ reaction on the other hand efficiently absorb light even at relatively low quantities (browning was observed immediately after introducing butenedial to basic conditions). The light absorptivity of the products can be attributed to the alkene bond they inherit from butenedial and potentially the presence of carbonyls. Therefore, it is suggested that other unsaturated dicarbonyls could lead to the production of light-absorbing compounds, although unsaturated compounds and basic pH are not expected to be relevant in atmospheric aerosols.





The third important difference is the rate and rate law for the dominant products of the dicarbonyl/NH$_X$ reactions. We
do not suggest that NH$_4^+$ is a catalyst (as a source of Bronsted acid) for butenedial/NH$_3$ reaction, which has been proposed by
previous studies for glyoxal (Nozière et al., 2009). While imidazole production is second order in glyoxal and NH$_3$ and
explicitly pH dependent, rendering it inefficient at low ambient concentrations (Yu et al., 2011), pyrrolinone production is
linearly dependent on butenedial and NH$_3$, which is similar to methylglyoxal (Sareen et al., 2010) and biacetyl (Grace et al.,
2020). This may not be surprising because in contrast to the two-carbon glyoxal, reaction between one butenedial and one NH$_3$
already results in a stable heterocycle. We suggest that dicarbonyls with a favorable separation of reactive aldehyde groups
can form heterocycles with bimolecular rate laws, which means they can occur even if ambient concentrations are low. They
could include four-carbon dicarbonyls (e.g., succinaldehyde, 4-oxopentanal, other unsaturated or saturated 1,4-dicarbonyls)
and phthalaldehyde, which can form five-membered rings, and five-carbon dicarbonyls (e.g., glutaraldehyde) that are capable
of forming pyridines and other six-membered heterocycles, as shown by Kampf et al. (2016). Electron poor dialdehydes with
longer carbon backbones (six-carbon or more) may also be able to produce stable products from self reactions. Notably,
butenedial reaction with NH$_3$ is much faster than for glyoxal, methylglyoxal, and biacetyl. Yu et al. (2011) showed that only
12% of the initial glyoxal in a 1 M glyoxal/1 M AS solution had been consumed over 5.5 months, whereas at comparable pH
we observed 11% removal of butenedial in a 0.4 M butenedial/0.2 M AS solution after 8 hours. Similarly, the lifetime of
methylglyoxal with respect to reaction with NH$_4^+$ in a 14 M AS solution is 29.8 hours (Sareen et al., 2010), however we
calculate a corresponding lifetime of 4.2 hours at pH 4, and 2.5 minutes at pH 6 for butenedial. The fast rate of butenedial/NH$_X$
reaction supports other work that has shown more rapid brown carbon formation from larger dicarbonyls than for α-dicarbonyls
(Kampf et al., 2016).

In sum, this work complements a previous study (Birdsall et al., 2019) to show that it is difficult to extrapolate the
physicochemical properties of reactive dicarbonyls from their α-dicarbonyl prototypes. Not only the reactive functional groups
affect reactivity but the structure of the carbon backbone as well. In the case of glyoxal, it is likely that its vicinal two aldehydes
cause its chemical and physical properties to be unique and dissimilar to the rest of the dialdehydes/dicarbonyls. Further studies
should be conducted with other more complex dicarbonyls to elucidate patterns in chemical behavior that are related to the
carbon skeleton.

### 4.2 Atmospheric implications

This work shows that the aqueous reaction of butenedial with and without NH$_X$ can form low-volatility chromophores that will
be retained in the condensed phase and absorb radiation. The results, especially the rapid reaction of butenedial with NH$_X$,
show that similarly reactive dicarbonyls could impact chemical composition and optical properties of particles, and thus
directly influence the human health and climate impacts of particles.

Butenedial was recently shown to have a gas-phase photochemical lifetime of 10-15 minutes due to photolysis
(Newland et al., 2019) although its high effective Henry's Law constant (6x10$^7$ M atm$^{-1}$, Birdsall et al., 2019) can allow
effective partitioning into the aqueous phase. Butenedial/OH$^-$ reactions are too slow at typical atmospheric pH values to



contribute significantly. It is, however, likely that condensed phase reaction of butenedial with $NH_X$ could regionally be important, specifically at close to neutral pH and high $NH_X$, such as in agricultural areas in India where $NH_3$ emissions are high (Kuttippurath et al., 2020) and rainwater is observed to be alkaline (Kulshrestha et al., 2001). At pH 6 and 4 M $NH_X$, the

lifetime of butenedial against reaction with $NH_X$ is only 18 minutes, even excluding enhancements from reaction with pyrrolinone and other accretion reactions or, in the atmosphere, other reactive organics. The results show that accretion reactions, and therefore the accumulation of chromophores, increase strongly with pH. Dimers and accretion products correspond to the vast majority of products (and quickly pull more butenedial out of solution) at slightly acidic or neutral pH. An important future research step is to refine the relationship between pH and oligomerization/accretion reaction rates.

355         We do not propose that butenedial alone contributes significant brown carbon. However, the reactivity of butenedial can be extrapolated to dicarbonyls for which condensed phase chemistry has not been studied. Such dicarbonyls are rarely measured but could be abundant. For example, 4-oxopentanal (a saturated 1,4-ketoaldehyde) was recorded at particulate concentrations averaging 62.7 ng m$^{-3}$ over a Japanese forest (Matsunaga et al., 2004); for a typical aerosol liquid water content of 1-10 mg m$^{-3}$, this corresponds to particulate 4-oxopentanal concentrations of approximately 6-60 mM, which was similar to

that of glyoxal and methylglyoxal. If 4-oxopentanal is representative of a range of dicarbonyls that react with $NH_X$ like butenedial, then the reaction could be fast enough that the sum of dicarbonyls may constitute a regional source of brown carbon in regions with high $NH_X$ and alkalinity. The vast majority of studies of the condensed-phase atmospheric chemistry of dicarbonyls have focused on glyoxal, methylglyoxal, and biacetyl, in part because they are abundant and commercially available. The fact that butenedial has much faster reaction rates of forming brown carbon that are first order indicates that

additional studies of larger dicarbonyls with hydrophobic moieties are needed, especially to further evaluate the role of dicarbonyls in the formation of brown carbon.

*Code and data availability.* The Python package pyrosolchem used as the kinetic model of droplet evaporation is available at https://github.com/jackattack1415/pyrosolchem (last access: 23 February 2021). Data used to generate paper figures are

available upon request.

*Author contributions*. JCH, AWB, and FNK designed the experiments. AWB, JCH, and GV performed the butenedial synthesis. JCH and JLC performed the NMR experiments. JCH performed the MS experiments. JCH analyzed the data and produced the model characterization. JCH prepared the paper with contributions from all co-authors.


*Competing interests*. The authors declare that they have no conflict of interest.

*Financial support*. This research has been supported by the National Science Foundation, Division of Chemistry (grant no. 1808084) and the Harvard University Faculty of Arts and Sciences Dean's Competitive Fund for Promising Scholarship.





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





**Table 1:** List of experiments performed, according to reaction. pH ranges and initial butenedial (BD) and $NH_X$ concentrations are provided.

| Reaction | $[BD]_0$ (M) | $[NH_X]_0$ (M) | pH |
|---|---|---|---|
| BD/$H_2O$ | 0.029-0.55 | 0 | 3 |
| BD/$OH^-$ | 0.2-0.3 | 0 | 8.8-10.4 |
| BD/$NH_X$ | 0.4-0.9 | 0.2-0.9 | 3.6-8.8 |


**Table 2:** Butenedial/$OH^-$ reaction, kinetic expression of the rate law, and corresponding estimates of the coefficients in the rate constant and their standard errors. $a_I$ and $a_{III}$ are well constrained. Although $a_{II}$ has a large standard error, it appears to not severely impact agreement between model and measurement (Figure 3). See Equation 1 for the expression of $k_1$ in terms of its coefficients, $a_I$, $a_{II}$, and $a_{III}$, and $OH^-$.

| Reaction | Kinetic expression | Estimated coefficients |
|---|---|---|
| (R1) $OH^-$ → products | $k_1$ [BD] | $a_I = 15.5 \pm 0.869$ $M^{-1}$ $s^{-1}$, $a_{II} = 64.6 \pm 4.00\times10^3$ $M^{-2}$ $s^{-1}$, $a_{III} = 1.61\times10^4 \pm 2.43\times10^3$ $M^{-1}$ |

**Table 3:** Reactions in the butenedial/$NH_X$ chemical mechanism, their rate laws as expressed in the mechanism, and corresponding estimates of the rate constants and their standard errors.

| Reaction | Kinetic expression | Estimated rate constant |
|---|---|---|
| (R2) | $k_2$ [BD] [$NH_3$] | $25.1 \pm 3.72$ $M^{-1}$ $min^{-1}$ |
| (R3) | $k_3$ [BD] [PR] [$OH^-$] | $6.36\times10^6 \pm 1.94\times10^6$ $M^{-2}$ $min^{-1}$ |
| (R4) | $k_4$ [BD] [AP] | $0.255 \pm 1.80\times10^{-2}$ $M^{-1}$ $min^{-1}$ |
| (R5) | $k_5$ [PR] [AP] | $0.460 \pm 3.98\times10^{-2}$ $M^{-1}$ $min^{-1}$ |
| (R6) | $k_6$ [BD-PR] [AP] | $0.172 \pm 2.18\times10^{-2}$ $M^{-1}$ $min^{-1}$ |



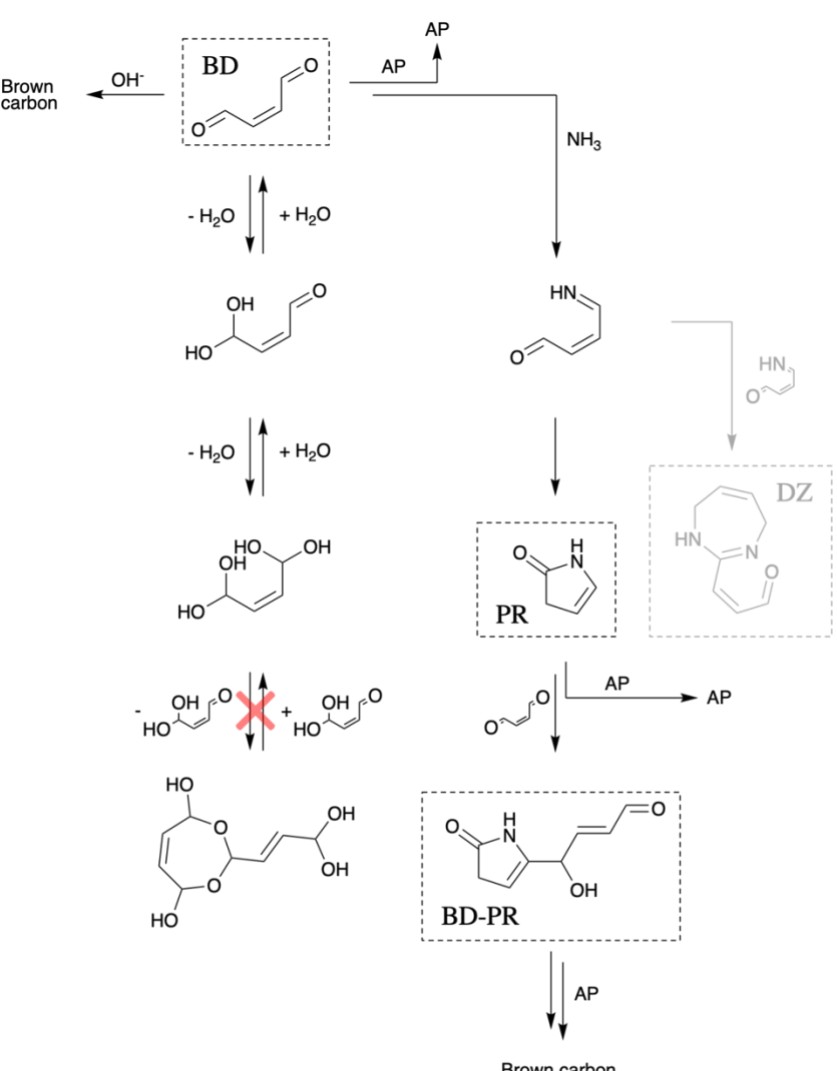

**Figure 1:** This chemical scheme is proposed for butenedial reactivity in an aqueous solution with $NH_X$. Reactions that are grayed out were not explicitly included in the kinetic mechanism. Butenedial/OH⁻ and butenedial/$NH_3$ reactions lead to brown carbon formation through accretion reactions. Accretion reactions with reactants, products, and accretion products (AP) are observed in the butenedial/$NH_3$ pathway. Acetal oligomer formation is not observed.





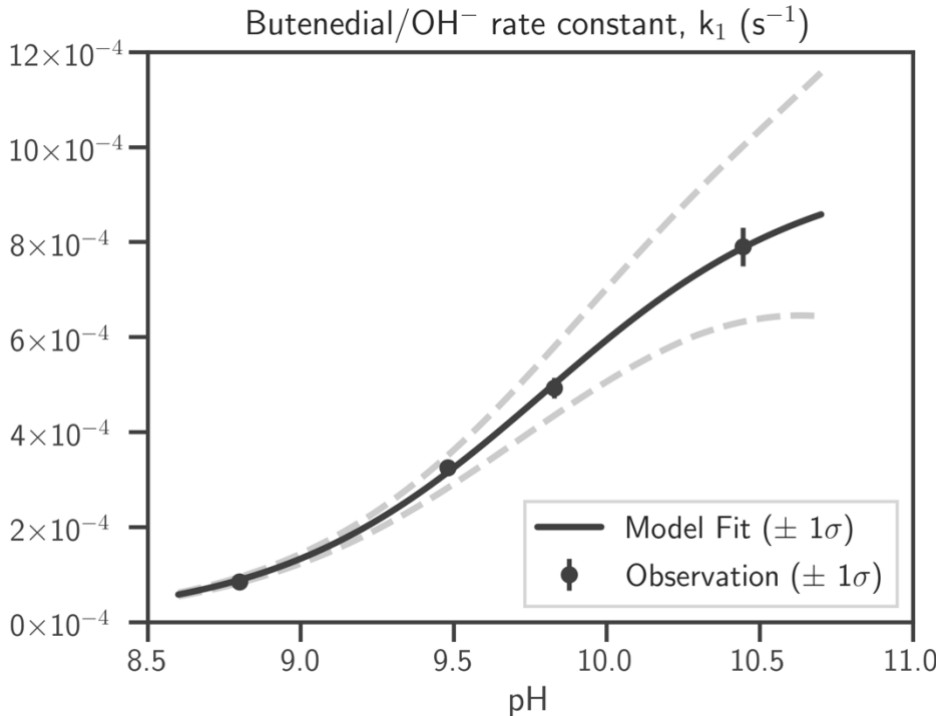

**Figure 2:** The butenedial/OH⁻ rate constant versus pH is shown. Observed $k_1$ are from kinetic fits to butenedial decay measured through four BD/OH⁻ experiments each held at constant pH. A fit to the empirical formulation of Fratzke et al. (1986) shows good agreement for all 550   observations.





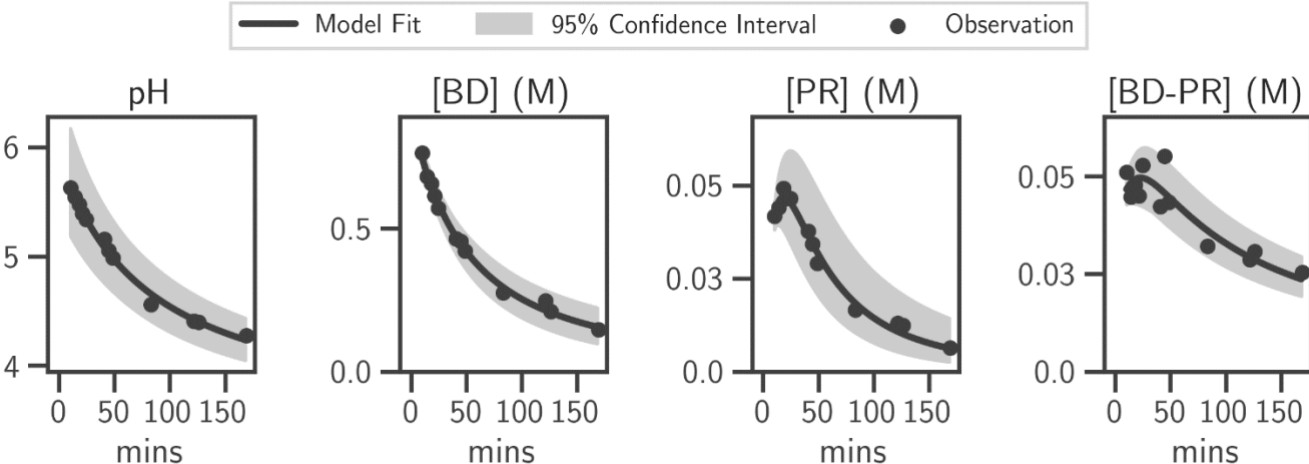

**Figure 3:** Comparison of measurement and model fit of 0.9 M BD/0.9 M NH$_X$ solution. Butenedial/NH$_X$ kinetic mechanism was fit to
measurements of this solution. The output (best model fit and 95% confidence interval) and observations are shown for all modeled species.
NH$_X$ was not measured explicitly but was assumed to be consumed at a 1:1 ratio with butenedial. pH was estimated empirically outside of
the model fit. Only the best model fit of pH was taken as input into the kinetic mechanism, although the 95% confidence interval is reported
for the empirical fitting of pH.





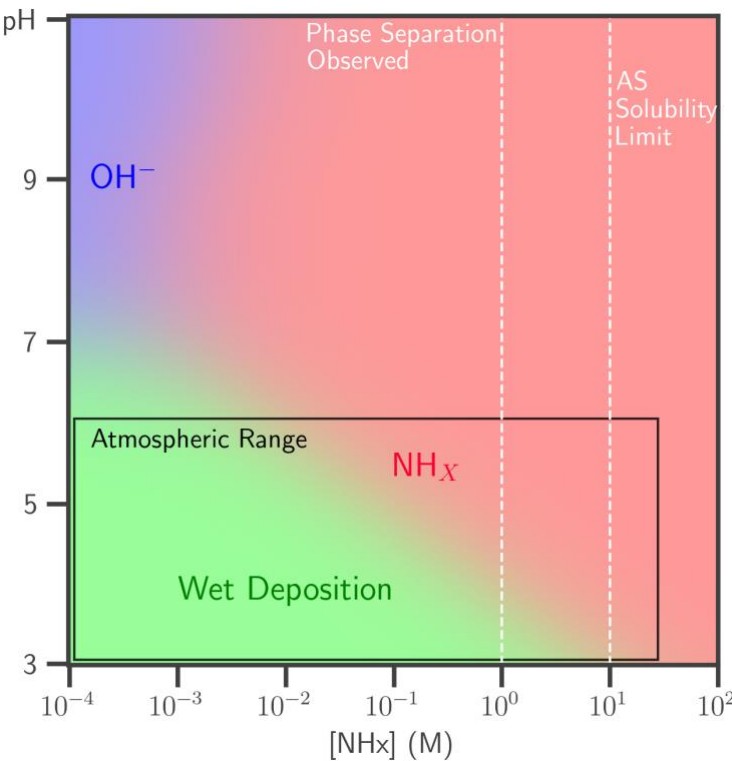

**Figure 4:** The dominant butenedial loss pathway, reaction with OH⁻ (blue) or NH$_X$ (red) and wet deposition (green), is shown as a function of pH and NH$_X$. AS is the NH$_X$ source. Loss via wet deposition is considered to have a one-week lifetime, typical of atmospheric particles. The range of pH (3-6) and NH$_X$ concentration (<28 M) relevant to the atmosphere is overlaid on the plot, as well as the NH$_X$ concentrations at which phase separation in the mixtures was observed and the AS solubility limit.