# Peer review of "Revisiting the reaction of dicarbonyls in aerosol proxy solutions containing ammonia: the case of butenedial"

_Atmospheric Chemistry and Physics, 2021_

## Author Comment (AC1)

10 May 2021

**Author Response for "Revisiting the reaction of dicarbonyls in aerosol proxy solutions containing ammonia: the case of butenedial" by Jack C. Hensley et al.**

We thank the anonymous referee for their thoughtful comments, which have helped improve the manuscript. Our replies are below (referee comment in bold face, response in normal face, manuscript indented with new content in italics, maintained content in normal face, and removed content in strike-through).

**This paper summarizes aqueous-phase reaction studies of butenedial as it reacts with OH⁻ (at high pH) and NH₃ / NH₄⁺ ("NHₓ," at neutral and acidic pH). In both cases, brown carbon is produced. Rates are compared with wet deposition as sinks for butenedial, and it is concluded that reaction with NHₓ can compete with wet deposition at mildly acidic pH or high aerosol-phase NHₓ concentrations. The atmospheric significance of butenedial should be better established at the beginning of the article, but in general, the work is extremely thorough and the conclusions are convincing.**

**Specific comments:**
**Line 41: The introduction contains many chemical justifications for studying aqueous-phase chemistry of butenedial, but this sentence is the only atmospheric justification given: "Larger, complex dicarbonyls are also thought to be important products of biomass burning and fossil fuel combustion (Arey et al., 2009; Aschmann et al., 2011, 2014; Gomez Alvarez et al., 2007, 2009; Volkamer et al., 2001; Yuan et al., 2017), but they have rarely been studied or quantified in the atmosphere. " It would be helpful if the authors could here address the question of whether butenedial has ever been reported in an aerosol field study or a chamber oxidation study. If not, why was it specifically selected? It would seem odd to study the chemistry of a reactant that has not yet been identified in the atmosphere or in a chamber study.**

The history of butenedial measurement in ambient air and chamber oxidation studies is further described on Line 97. This account has been extended in the manuscript to better motivate studies of butenedial, as suggested by the reviewer:

>  *Butenedial is an unsaturated 1,4-dialdehyde known to be a major oxidation product of atmospherically abundant aromatics and furans (Bierbach et al., 1994; Coggon et al., 2019; Stockwell et al., 2015; Strollo and Ziemann, 2013; Volkamer et al., 2001; Raoult et al., 2004; Müller et al., 2016). Laboratory studies of OH oxidation of precursor compounds have recorded butenedial yields of 10.3% (Berndt and Böge, 2006) and 16% (Gómez Alvarez et al., 2007) from benzene; 13% (Gómez Alvarez et al., 2007) and 11-32% (Arey et al., 2009) from toluene; 10-29% (Arey et al., 2009) from o-xylene; and 99% (Gómez Alvarez et al., 2009) and 75% from furan (Aschmann et al., 2014). Butenedial and other dicarbonyls were detected in ambient air samples with elevated loading of aromatic hydrocarbons (Obermeyer et al., 2009). To the best of our knowledge, butenedial has not yet been quantitatively measured in field studies, in part due to the aforementioned challenges of measuring dicarbonyls.*

**Line 51: The ease of using bulk-phase methods to study aerosol-phase chemistry is described here, but the drawbacks of using only these methods should also be described. These include lack of sites for surface chemistry to take place.**

10 May 2021

In an upcoming manuscript, we contrast butenedial/NH$_X$ reaction in bulk solutions and levitated particles, which considers the limitations of bulk solution studies when extrapolated to atmospheric aerosol. A sentence has been added in the manuscript to address the major limitations, starting on Line 51:

> Bulk solutions can mimic the aqueous phase of actual atmospheric particles, albeit at lower ionic strength *and reagent concentrations and without surface effects, which may influence particle phase chemistry (Yan et al., 2016; McNeill, 2015). The extrapolation of chemistry quantified in bulk solutions to atmospheric particles is evaluated in another study (Hensley et al., 2021, in prep). Bulk solutions have the additional* advantage […]

**Line 21, 61-64, 317 and 320: According to Noziere et al., 2009, and Sedehi et al., 2013, the quadratic dependence of the rate on glyoxal concentrations is only observed at high concentrations, and switches to 1$^{st}$ order at concentrations below the molar range. The argument (made three times in the manuscript and referenced as a key difference from glyoxal in the abstract) that the quadratic rate law contributes to the reaction shutting off at low concentrations is therefore not sound. The paragraph in the discussion should be rewritten to avoid a reliance on this idea. But I agree with the rest of the statement at line 62.**

Thank you for this point. The manuscript has been updated such that comparison of butenedial and glyoxal reaction with NH$_X$ does not rely on a difference of reaction order at low concentrations:

Lines 63-69:

> Yu et al. (2011) and Kampf et al. (2012) demonstrate that imidazole formation from glyoxal/NH$_X$ follows a rate law of the form k[GL]$^2$[NH$_4^+$][NH$_3$], and is as such fastest at neutral to basic pH (Maxut, 2015). *Noziere et al. (2009) suggest that glyoxal/NH$_X$ is second order at low reactant concentrations, where imine formation is the rate-limiting step (Sedehi et al., 2013).* Although imidazoles have also been detected in chamber experiments of deliquesced AS aerosol, the reaction is observed to be too slow at typical dicarbonyl concentrations and aerosol pH and NH$_X$ to affect chemical composition,  (Galloway et al., 2009; Yu et al., 2011).

Lines 341-342:

> While imidazole production is second order in glyoxal and NH$_3$ and explicitly pH dependent,  (Yu et al., 2011), pyrrolinone production is linearly dependent on butenedial and NH$_3$,

Lines 344-345:

> We suggest that dicarbonyls […] can form heterocycles with bimolecular rate laws,

**Line 108: The manuscript states that no overlapping products were observed between OH$^-$ and NH$_x$ experiments. However, the OH$^-$ reaction products were detected using negative ion mode, and the NH$_x$ reaction products were detected using positive ion mode on**

**a different mass spectrometer. Were the two reactions tested using the other mass spectrometric methods to screen for these overlapping products? I did not find a clear answer to this question in the manuscript or supplement.**

We find that butenedial/OH$^-$ reaction is slow at pH < 9 (0.9 M butenedial/0.45 M AS has solution pH < 6) and butenedial/NH$_X$ reaction is negligible in butenedial/OH$^-$ experiments, which did not have a source of NH$_X$. This demonstrates that these reactions were studied in isolation of one another. We observed that product spectra in NMR measurements of butenedial/OH$^-$ and butenedial/NH$_X$ reaction mixtures are not the same, although this was a qualitative verification and is omitted from the manuscript. Additionally, we did not observe evidence of butenedial/NH$_X$ reaction products in butenedial/OH$^-$ reaction mixtures with LC-MS operated in positive mode.

The manuscript is updated to explicitly articulate the above on lines 117-119:

> As is  *shown* later, reaction with OH$^-$ was found to be negligible at the pH and NH$_X$ conditions of the butenedial/NH$_X$ reaction studies, and vice versa, *indicating that butenedial/OH$^-$ and butenedial/NH$_X$ reactions were studied in isolation of one another*.

**Technical Corrections:**
**Line 148: The manuscript states that 0.5 M sodium carbonate (Na$_2$CO$_3$) – sodium bicarbonate (NaHCO$_3$) buffer was added to every reaction solution with ammonium. Table 1 states that a pH range of 3 to 9 was studied, but the carbonate buffer could only be used at the upper end of this range. How was pH experimentally set to values less than 8?**

In butenedial aqueous solutions with NH$_X$, initial pH is determined by: (1) residual acetic acid from butenedial synthesis, (2) ammonium sulfate (AS), and (3) carbonate buffer system, if added. Without addition of carbonate buffer, the starting pH of solution is >3-4, as was the case for solutions with initial composition 0.4 M butenedial/0.2 M AS (Figure S22).

The reviewer refers to solutions with initial composition 0.9 M butenedial/0.45 M AS, with 0.5 M carbonate buffer added. The starting solution pH ~6 reflects the consumption of some acidity by the added buffer. As seen in Figure 3, pH decreases during reaction, presumably due to a reaction byproduct, as was demonstrated for glyoxal/AS reaction (Yu et al., 2011). The influence of the identified factors on solution pH for the 0.9 M butenedial/0.45 M AS experiments is now specified in Section 2.3.2 on lines 204-206:

> *pH was affected by residual acetic acid from butenedial synthesis, ammonium sulfate, addition of carbonate buffer, and production of acid during reaction.* pH was estimated with an empirical formulation that agreed closely with measurements (Figure S19).

**Line 182: This sentence refers to a fitting procedure described in section 2.3.1, but the sentence is part of section 2.3.1. A typo?**

Section 2.3.2 has now been appropriately labeled on line 200.

10 May 2021

**2.3.1** *2.3.2* **Butenedial/NH$_X$ reaction**

10 May 2021

**References**

[revised manuscript text omitted]

---

## Author Comment (AC2)

10 May 2021

**Author Response for "Revisiting the reaction of dicarbonyls in aerosol proxy solutions containing ammonia: the case of butenedial" by Jack C. Hensley et al.**

We thank the anonymous referee for their thoughtful comments, which have helped improve the manuscript. Our replies are below (referee comment in bold face, response in normal face, manuscript indented with new content in italics, maintained content in normal face, and removed content in strike-through).

**General comments:**
**This manuscript describes results from an analysis of the reactions that can occur for butenedial in aqueous solution as a function of pH and as a function of NHx concentration. The chemical changes were tracked with NMR combined with some MS to help identify products and a chemical scheme along with reaction kinetics are provided. Overall, this work demonstrates the need for additional studies on different types of dicarbonyls that are atmospherically relevant as the behavior of butenedial does not follow what would be predicted based on prior studies of a-dicarbonyls like glyoxal and methylglyoxal. This is a well written and clear study that builds on prior work. I would recommend acceptance in ACP after the following minor comments are addressed:**

**Minor comments:**
   1. **The accretion products from betenedial/OH- were observed to be brown immediately, and a portion of the MS is provided for the samples (figure S4). Were there any nitrogen containing peaks observed to form in this sample in the other mass ranges? I am concerned about trace ammonia from the room since very small concentrations would be needed if the chromophores have a large absorption cross section.**

Thank you for this comment. To confirm that butenedial/OH$^-$ products are light-absorbing, new measurements with high resolution LC-MS-UV/Vis were taken of butenedial/OH$^-$ reaction mixtures. As shown in Figure S5, reprinted below, we observe evidence for proposed hydroxy acid oligomer products, with unambiguous molecular formula $C_{4n}H_{4n+1}O_{2n+1}^-$ (deprotonated under pH conditions of mixture). As shown in Figure S6, reprinted below, these products absorb in the 300-450 nm wavelength range. Additionally, we did not observe evidence of butenedial/NH$_X$ reaction products in butenedial/OH$^-$ reaction mixtures with LC-MS operated in positive mode.

Figure S5.

[Figure]

Figure S6.

[Figure]

**2. The concentration ranges here were reasonably high, what do these concentration ranges correspond to in the atmosphere? There is some discussion about rainwater in India, given the much lower concentrations for organics that can be found in rainwater, would the authors expect to see the same types of chemistry?**

Butenedial concentrations are unknown in the atmosphere, as to the best of our knowledge, butenedial has not yet been quantitatively measured. The measurements by Matsunaga et al. (2004) for 4-oxopentanal, a saturated 1,4-ketoaldehyde, suggest that larger dicarbonyls could be as abundant as 1,2-dicarbonyls, with concentrations in aerosol ranging from 6-60 mM. Such concentrations are 1-2 orders of magnitude less than those we used in this study. Unlike for

glyoxal, we do not find evidence for a dependence on butenedial concentration in the reaction rate law. As Reviewer 1 pointed out, others have suggested that glyoxal may switch to a first order dependence at low concentrations, as under these conditions, the bimolecular reaction of glyoxal and $NH_X$ may be the rate limiting step (Nozière et al., 2009; Sedehi et al., 2013). Pyrrolinone formation is shown to be first order with respect to butenedial regardless of whether imine formation or ring-closure is the rate limiting step.

Research on the pH of rainwater in India is alkaline due to dissolution of $Ca^{2+}$-rich aerosols (Kulshrestha et al., 2001). We therefore take this as a proxy for geographical location in which pH conditions could be basic enough to favor butenedial/$NH_X$ reaction. The manuscript is updated to make this point more clearly. It also includes the North China Plain as another similar region with high $NH_X$ and pH, lines 371-374:

> It is, however, likely that condensed phase reaction of butenedial with $NH_X$ could regionally be important, specifically at close to neutral pH and high $NH_X$, such as in agricultural areas in India *or the North China Plain* where $NH_3$ emissions are high (Kuttippurath et al., 2020*; Zhang et al., 2010*) and  *aerosol may be alkaline* (Kulshrestha et al., 2001; *Tao et al., 2020*).

3. **On page 7, it is stated that the 1H-NMR spectra shows a buildup of signal in the baseline which increases and spreads out with respect to the chemical shift over time. With the data overlaid and colored the way it is, this is very difficult to see in the figure. Also in Figure S5, there is a note that there are two expanded regions, but these are not shown in the figure.**

Figure S7 (previously Figure S5), reproduced below, contains the described expanded regions to improve visibility of the buildup of signal in the baseline, and is reprinted below. Additionally, we provide new evidence of the accretion reactions that cause the observed build-up in the baseline through high-precision LC-MS measurements of hydroxycrotonic acid oligomers in butenedial/$OH^-$ reaction mixtures (see Figure S5 above).

10 May 2021

Figure S7.

[Figure]

4.  **The numerical values for the last few supplemental figures appear to be off in the manuscript (there is no Figure S16).**

References to the supplemental section now have been checked to ensure that they match the correct figures.

10 May 2021

**References**

[revised manuscript text omitted]

---

## Author Comment (AC3)

10 May 2021

**Author Response for "Revisiting the reaction of dicarbonyls in aerosol proxy solutions containing ammonia: the case of butenedial" by Jack C. Hensley et al.**

We thank the anonymous referee for their thoughtful comments, which have helped improve the manuscript. Our replies are below (referee comment in bold face, response in normal face, manuscript indented with new content in italics, maintained content in normal face, and removed content in strike-through).

**Hensley and coworkers studied the reaction of butenedial, which has been observed in lab studies and ambient air, with OH- and water in the absence of NHx and in ammonium sulfate (AS) solutions. Products and rates were monitored with 1H HMR. The butenedial was synthesized. LC-TOF-MS data were also taken to analyze products. This is a solid study. Generally I found the discussion of the chemistry, or analytical assignments, could have been more specific and clear. I have the following comments that should be addressed prior to publication.**

**Can "kinetic mechanism" be changed into something more clear like "kinetic modeling mechanism"?**

We have changed the terminology from "kinetic mechanism" to "model kinetic mechanism" throughout the text. Any mention of "model kinetic mechanism" explicitly refers to the set of differential equations that govern the kinetics of the chemistry. To distinguish between the chemical speciation and kinetics, three mentions of "chemical mechanism" have been removed and replaced with "chemical scheme" and "model kinetic mechanism."

**Can the authors provide more information into how products such as pyrrolinone were assigned? More discussion is need to support major reaction route and products in Figure 1. How were products confirmed or ruled out based on NMR shifts? I see that the SI has some assignments, but the text should refer to the SI when these assignments are called on, and it's not clear how these assignments were made and no citations to analogous compounds in the literature. It's a good idea to also discuss any limitations when it comes to unambiguously assigning structure from NMR shifts in a complex mixture and or from m/z.**

Thank you for this comment. The suggestions have strengthened our paper. As we did not have standards for proposed products against which to compare NMR and MS spectra, the proposed products were identified in situ with a combination of NMR and MS. TOF-MS with unit mass precision and LC-MS with <5 ppm mass precision were used to identify unambiguous molecular formula of reactant and reaction products. We recognize that unambiguous assignment of NMR shifts to specific molecules in a complex mixture can be tentative at best, and the language throughout the text now reflects this. However, as we describe below, the proposed products of reaction are consistent with the exact chemical formulae determined by MS, the expected NMR assignment as well as well-established understanding of organic chemistry of this type of chemical systems (e.g., glyoxal/NH$_X$).

10 May 2021

As is discussed in Section 2.2.2 of the main text, the species identification methodology was as follows: (1) determine molecular formula with MS measurement, (2) group NMR-identified protons based on integer signal, maintained through time, that are temporally consistent with speices measured with MS, (3) derive proposed proton assignment in NMR measurements to determine structures of proposed products.

1. The two proposed major products of butenedial/$NH_X$ reaction, $C_4H_5NO$ and $C_8H_7NO_2$, were identified with the observed m/z channels of formed products in butenedial/AS solutions. With new LC-MS measurements of both butenedial/$NH_X$ and butenedial/$OH^-$ reaction mixtures at high mass precision, these molecular formulae were unambiguous (Figure S12-S13, reproduced below, which also includes an accretion product of three butenedial units). Based on the molecular formulae, the reactant products have the same degree of saturation as their parent compounds.

Figure S12.

[Figure]

Figure S13.

[Figure]

2. As is shown in Table S5, excluding butenedial, we observe two groups of protons with integer signal that is maintained through the progress of reaction. They are, in δ (ppm) [number of protons], Group 1: 6.55 [1], 5.92 [1], 3.37 [2] and Group 2: 6.29 [1], 6.03 [1], 5.97 [1], 5.65 [1], 5.43 [1], 3.41 [2]. In a new figure, Figure S15, reproduced below, temporal agreement between NMR and MS measurements indicate that "Group 1" corresponds to $C_4H_5NO$ and "Group 2" corresponds to $C_8H_7NO_2$.

Figure S15.

[Figure]

3. Pyrrolinone: the specific structure of $C_4H_5NO$ is proposed through analogous reactions and consistency with expected NMR protons. Following work on glyoxal/$NH_X$ reactions (Yu, Kampf), butenedial reaction with $NH_X$ converts an aldehyde into an imine. The proposed imine has molecular formula $C_4H_5NO$. However, this molecular structure cannot explain the observed upfield protons (two at 3.37 ppm). We suggest that ring closure through the Paul Knorr mechanism results in a pyrrole. Subsequent rearrangement, which has been observed for hydroxypyrroles, leads to the proposed pyrrolinone product. We suggest the two upfield protons at 3.37 ppm belong to the methylene group, while the downfield protons at 6.55 and 5.92 ppm are vinylic. The proton with signal at 6.55 ppm is assumed to be attached to the carbon adjacent to the NH group, while the proton with signal at 5.92 is assumed to be attached to the other vinyl carbon. Agreement was verified with NMR prediction software (https://www.nmrdb.org/new_predictor/index.shtml?v=v2.121.0, last accessed: 2021 April 11) (Banfi and Patiny, 2008).

Butenedial-pyrrolinone "dimer": To the best of our knowledge, butenedial-pyrrolinone (BD-PR) "dimer" has not be studied and is assigned tentatively in this work. Given that (1) pyrrolinone is a likely product of reaction, (2) aldehyde-pyrrole condensation reactions are well-established to take place (source), (3) the proposed butenedial-pyrrolinone "dimer" has molecular formula $C_8H_7NO_2$ (therefore agrees with observed

product at m/z 150 consisting of two butenedial building blocks and one $NH_X$), and (4) glyoxal forms similar "dimers" with imidazole, the product of its reaction with $NH_X$, we propose that BD-PR "dimer" the second observed species in NMR. The molecule is assigned the "Group 2" protons. Tentative assignments are based off of assignments for butenedial and pyrrolinone. Downfield protons (6.29, 6.03, and 5.97 ppm) are assigned according to the carbon skeleton derived from the butenedial part of the "dimer," furthest from the linkage. The methylene protons are assigned to 3.41 ppm. According to prediction software, the vinylic proton on the ring should be slightly downfield (5.65 ppm) and the proton of the carbon with a single hydroxyl group should be slightly upfield (5.43 ppm) (Banfi and Patiny, 2008). Agreement was verified with NMR prediction software (Banfi and Patiny, 2008).

There may be other minor products, such as the diazepine, produced from the reaction, although they are not expected to be significant enough to warrant inclusion in the mechanism. Any major species that could be undetectable with one technique (due to e.g., low proton affinity in the case of MS and spectral interference in the case of NMR) should be observed with the other. Our new LC-MS-UV/Vis studies of the products also show clear evidence for an accretion product consisting of three butenedial and two $NH_X$, providing further evidence that this accretion process is active.

Section 2.2.2 has been rewritten to be more explicit about how products of BD/$NH_X$ reactions were determined, as is shown here.

0.9 M butenedial/0.45 M AS (VWR, > 99%) mixtures were prepared in water and $D_2O$ with the internal standards PEG-6 or DMS and 0.5 M sodium carbonate ($Na_2CO_3$) – sodium bicarbonate ($NaHCO_3$) buffer. The solution immediately turned orange brown *(Figure S16)*. After 20 min of reaction, mass spectra of the mixtures indicated nitrogen-containing products with signals at m/z 84, 149, 150, and 168, assumed to be adducts with $H^+$ *(Figure S11)*. The most reasonable chemical formulas of these products were $C_4H_5NO$ (83 Da), $C_8H_8N_2O$ (148 Da), $C_8H_7NO_2$ (149 Da), and $C_8H_9NO_3$ (167 Da). $C_8H_9NO_3$ was the parent molecule for the $C_8H_7NO_2$ fragment. *$C_4H_5NO$ (83 Da), $C_8H_9NO_3$ (167 Da), and $C_8H_9NO_3$ (251 Da) were observed unambiguously with high-resolution LC-MS measurement of an equivalent solution (Figure S12).*

The 1H-NMR spectra (Figure S14) showed *two* distinct groups of quantitative related signals that had similar temporal behavior (Table S5). Each group of peaks whose quantitative signal strength behaved as integers and had the same temporal behavior was presumed to arise from a single compound. *One group was assigned to $C_4H_5NO$ and the other to $C_8H_9NO_3$ according to agreement in chemical evolution between MS and NMR measurements (Figure S15).* A molecular structure was proposed for each cluster of peaks and the molecular formulas mentioned above, *according to NMR peak assignments and analogous reactions (see SI Section 2.4, including Figures S9-S10).* The inferred products were as follows: 2-pyrrolinone (pyrrolinone, PR, $C_4H_5NO$), and a butenedial-pyrrolinone "dimer" (BD-PR, $C_8H_9NO_3$). We propose that 2-butenal-1,3-diazepine (diazepine, DZ, $C_8H_8N_2O$) is a minor product that is observable with MS but was not detected with the less sensitive NMR. The growth of broad peaks embedded in the baseline suggested substantial formation of accretion products *(Figure S14). Additionally, high-resolution LC-MS-UV/Vis measurements suggested evidence of pyrrolinone, butenedial-pyrrolinone "dimer," and a "trimer" formed from addition of butenedial and $NH_X$ to the "dimer" (Figures S12-S13). In particular, the proposed "dimer" and "trimer" are strongly π-conjugated and light-absorbing (Figures S12-S13). As such, accretion products composed of butenedial, NHx, and pyrrolinone* could explain the dark color of the solution *(Figure S17)*.

10 May 2021

**Some of the figures/tables in the SI are not referred to in the main text. For example, when the authors mention the solutions turn brown they should probably refer to the figures they provided in the SI, otherwise the reader does not know to look.**

We have included several references to the SI throughout the main text to guide the reader to important figures.

**92 What is the concentration of butenedial in ambient air? Is it observed in the gas or condensed phase? How do these concentrations compare with the initial concentrations the authors chose for this work, and if they are very different, please discuss how this work can extrapolate to the ambient environment.**

See response to first comment of Reviewer 1. Butenedial has not been measured quantitatively in ambient air, to the best of our knowledge.

**99 "react with OH-" is stated multiple times, without discussion about how. Please summarize what is known about how OH- reacts with the moieties of interest, and please discuss the specific mechanism. Same thing with NHx.**

Reaction with NH$_X$.
Thank you for this suggestion. As is discussed in Section 2.4 of the SI, carbonyl/NH$_3$ reactions is proposed to occur through well-established Paul Knorr synthesis. We have included new figures with movement of electrons for butenedial/NH$_3$ reactions (Figures S9-S10, shown below). A new reference to this description is on Line 175 of the main text.

Figure S9.

As shown, reaction is proposed to begin with NH$_3$ nucleophilic attack of a protonated carbonyl, which dehydrates, deprotonates, and forms an imine intermediate, as has been shown for, e.g., glyoxal (Nozière et al., 2009; Yu et al., 2011; Laskin et al., 2015). We propose that the imine undergoes ring closure in the case of butenedial, which would form a reactive hydroxypyrrole, and under acidic to slightly acidic conditions, is expected to tautomerize to the stable pyrrolinone form.

Figure S10.

As is discussed in Section 2.4 of the SI, pyrroles are known to tautomerize depending on pH (Capon, 1989). We propose that the OH⁻ dependence of reaction R3 arises from this pH-dependent tautomerization. Aldehyde-pyrrole condensation is well known through, for example, extensive study on the synthesis of polyphorins (Koelsch and Richter, 1935). As shown in Figure, hydroxypyrrole is proposed to undergo electrophilic substitution reaction with butenedial, which typically for heterocycles occurs at the second position (adjacent to NH group). The ring may tautomerize again. The reaction is analogous to glyoxal-imidazole accretion reactions that are also known to occur (Yu et al., 2011; Kampf et al., 2012).

Reaction with OH⁻.
Carbonyl/OH⁻ reactions tend to produce disproportionation products through the well-established Cannizzaro mechanism (source). Disproportionation reactions have been shown for many other dicarbonyls, including glyoxal, methylglyoxal, and phenolglyoxal (Fratzke and Reilly, 1986). Such reactions begin with OH⁻ nucleophilic attack of a carbonyl, and through subsequent hydride ion transfer to an adjacent carbonyl, the hydride donor carbonyl is oxidized to a carboxylate and the other carbonyl is reduced to an alcohol. As mentioned in the text, in the case of glyoxal, disproportionation produces the hydroxy acid, glycolic acid (Fratzke and Reilly, 1986).

The corresponding hydroxy acid for butenedial/OH⁻ reaction is γ-hydroxycrotonic acid. The molecular formula of γ-hydroxycrotonic acid and several of its oligomer products (up to the 11-mer) were observed with new LC-MS-UV/Vis measurements, as shown in Figures S5-S6, reproduced below. This suggests that disproportionation reactions take place for butenedial/OH⁻. We have included new figures with movement of electrons for butenedial/OH⁻ reactions (Figures S4, shown below). This information now has a reference in Section 2.1 of the main text.

Figure S5.

[Figure]

Figure S6.

[Figure]

Figure S4.

10 May 2021

**149 The mass spectra show a number of odd peaks. Did the authors rule out 2N products and how did they confirm 0N products? It would be good to specifically address whether this system is anticipated to form 2N cyclic compounds like in other dialdehyde systems. What is the mass precision after calibration in order to differentiate between 0-2N? (the assigned mass and peak shown in S4 are roughly 5 ppm off)**

Analysis of butenedial/$NH_X$ mixtures was performed with TOF-MS, with unit mass precision. As shown in previous work with the same TOF-MS, the internal standard, hexaethylene glycol, has a parent ion at m/z 283 and with fragments at m/z 63 + 44 n and occasionally clustered with water molecules (Birdsall et al., 2019, 2018). The only 0N species that was confirmed was butenedial, which we have measured previously at the m/z 85 channel (Birdsall et al., 2019). The only 2N species that could be at the m/z 85 channel (with a four-carbon backbone) is $C_4H_8N_2$, which would require saturation of the carbon backbone, and is not likely. As mentioned on Line 157, a diazepine is a 2N product that was observable with MS although it was not detected with NMR. This suggests that production of pyrrolinone through ring-closure of the imine is favorable compared to the "dimerization" reaction that would produce the diazepine, which would be analogous to imidazole formation. Diazepine is therefore a minor product of reaction.

Additional high-resolution measurements were performed with LC-MS-UV/Vis of both butenedial/$NH_X$ and butenedial/$OH^-$ reaction mixtures to determine unambiguous molecular formula. As shown in Figure S5 and S11 (reproduced in comments above and below), mass precision was <5 ppm. As mentioned in previous comments, in butenedial/$OH^-$ reaction mixtures, we observed evidence of a hydroxy acid (without N) and its oligomers. Additionally, LC-MS of butenedial/$OH^-$ reaction mixtures in positive mode did not show evidence of butenedial/$NH_X$ products.

With LC-MS-UV/Vis of butenedial/$NH_X$ reaction mixtures, we additionally confirm the previously determined molecular formulae of butenedial/$NH_X$ products.

**193-200 Can the authors provide citations for any of these reactions that they are suggesting, such as OH- reacting with BD to form a hydroxyacid (and by which pathway, i.e., addition/abstraction and where) and then ultimately leads to oligomeric light absorbing products (again, by which pathways). And what do the oligomeric light absorbing products look like?**

As mentioned in comment to Line 99 above and in the text, we reference the work of Fratzke & Reilly (1986), which has characterized glyoxal reaction with OH- and forms the chemical basis

for the proposed butenedial/OH⁻ reaction. Butenedial/OH⁻ reaction has not been studied previously, to the best of our knowledge. See response to the comment on Line 99 above for more information about the initial reaction pathway, as well as a proposed product of reaction, hydroxycrotonic acid. Hydroxy acids are known to oligomerize through condensation reactions (as is the case for lactic acid and glycolic acid).

With new LC-MS-UV/Vis measurements previously described, we observe evidence of hydroxycrotonic acid and its oligomers formed in butenedial/OH⁻ reaction mixtures (Figure S5, see comment 99 above). As shown in Figure S6 (reprinted in response to comment 99), hydroxycrotonic acid and its oligomers are capable of absorbing light in the 300-450 nm range, as is typical for brown carbon (e.g., Laskin, 2015). This strongly supports evidence for accretion products due to the build-up of material in the baseline of the NMR spectrum, which with conjugated $\pi$ bonds, is known to enhance the absorptivity.

**236 Reaction R3 is an example of an accretion product given in this work. Its assignment was shown in Figure S6/S8 and associated discussion. The text should reference the carbon/proton assignments in S6, and discuss how those assignments were made (including any NMR reference tables used and for which proxy molecules). Also instead of "produces" it should say "proposed to produce" because these are only tentative assignments after all. Please insert a few sentences to discuss the specific mechanism by which this product can form, via reaction R3 with OH-.**

Thank you for these points regarding butenedial/NH$_X$ accretion products. With new LC-MS-UV-Vis measurements of butenedial/NH$_X$ reaction mixtures, we observe unambiguous evidence for this "dimer" as well as a "trimer," composed of three butenedial building blocks and two NH$_X$, formed in solution (Figure S11, reproduced below). The strong signal intensity of both "dimer" and "trimer" suggests that accretion products, composed of butenedial and pyrrolinone, are substantial. We now rely more strongly on this evidence in the methods section of the text and in the SI than on the tentative NMR assignments, which as Reviewer 3 mentions previously, are difficult in complex reaction mixtures. Specific discussion of the butenedial-pyrrolinone "dimer" formation is in the response to the comment of line 99, is also in Section 2.4 of the SI, and now referenced in the main text, following the comment of line 99. Discussion of the NMR assignments is in the author response to Reviewer 3's second comment.

Figure S11.

10 May 2021

[Figure]

**Even though the solutions look visibly brown, the authors did use a lot of carbon material initially. What is the mass absorption coefficient (MAC) of these reactions and how do they compare to other brown carbon (e.g., from Updyke 2012)?**

The mass absorption coefficients were not measured for butenedial/NH$_X$ products in this study, although they would be important for accurate knowledge of the climate implications of the chemistry.

**320 clarify "favorable separation"**

Made more specific in the manuscript, see Line 344-345:

> We suggest that dicarbonyls with  *at least two carbons between carbonyl groups* can form heterocycles with bimolecular rate laws.

**345 I see that the authors considered the wet deposition lifetime of aerosols (~ 1 week) from Seinfeld and Pandis. I'd also suggest to calculate the wet deposition lifetime of the molecule based on equation 12 in this work (https://acp.copernicus.org/preprints/acp-2021-137/) to better motivate the importance of aqueous partitioning for this compound compared to its gas phase photolysis.**

As mentioned on Lines 369 of the text, butenedial has a very large Henry's law constant (~6×10$^7$ M atm$^{-1}$), indicating that it is highly soluble in water, much more so than methylglyoxal and glyoxal. Therefore, it is expected that atmospheric butenedial readily partitions to aqueous aerosol. We provide lifetimes of reactive loss for butenedial in particles, which as shown in the text, is 18 minutes for particles with pH 6 and 4 M NH$_X$. Thus, under high pH and NH$_X$ conditions, aqueous reaction is capable of competing with fast gas-phase photolysis, which has a first-order lifetime of 15 min (Newland et al., 2019).

The provided link is directed to this preprint. To determine time scales of mass transfer from gas to particle, following Maxwellian Flux (Seinfeld and Pandis, 2016), Zaveri et al. (2014) derive the following gas-particle partitioning flux for a species i:

$$\frac{dA_i}{dt} = \frac{3k_{g,i}}{r}\left(c_{g,i} - \frac{A_i}{\sum_j A_j}\, c_{g,i}^*\right) - k_r A_i$$

Where $A_i$ is the particle concentration of species i in mol cm$^{-3}$, $\sum_j A_j$ is the total molar concentration of all species in the particle in mol cm$^{-3}$, r is the particle radius, $k_{g,i}$ is the gas mass-transfer coefficient of species i in s$^{-1}$, $c_{g,i}$ is the gas-phase concentration of species i in mol cm$^{-3}$, $c_g^*$ is the effective saturation vapor concentration of species i in mol cm$^{-3}$, and $k_r$ is the first order rate constant of bulk phase reaction. Calculation of mass transfer timescales requires knowledge of the gas-phase concentration of butenedial as well as surface uptake coefficients that control $k_{g,i}$, neither of which are measured.

**330 the lifetimes decrease with increasing pH. Can the authors discuss relevance of this reaction to aerosol water, which tends to be acidic (such that BD would have lifetimes of > 4 h, and would photolyze before that) but would be the locations where one would find higher NH4+. Comparatively pH 6 might be cloud droplet range, but then ionic strengths are low.**

In Section 4.2, we discuss regions where high NH$_3$ and high pH coincide. Although aerosol pH is indeed typically acidic, certain locations, such as the North China Plain (NCP) or northern India, tend to have aerosol that contains alkaline components, high aerosol water content, and elevated NH$_3$. Recently, it was shown that near-surface aerosol pH was typically 4.4-5.7, and in some locations could consistently be > 6, in the NCP (Tao et al., 2020). As is mentioned in the reviewer comment, clouds have similarly high pH, but reactions in clouds may not be as relevant as reagent concentrations would be substantially reduced compared to aerosol. A sentence has been added to the manuscript to specify aerosol pH from this recent study at Lines 374-375:

[revised manuscript text omitted]